# The effectiveness of peer-support for people living with HIV: A systematic review and meta-analysis

Rigmor C. Berg[1,2]*, Samantha Page[3], Anita Øgård-Repål[4]

**1** Division for Health Services, Norwegian Institute of Public Health, Oslo, Norway, **2** Institute of Community Medicine, University of Tromsø, Tromsø, Norway, **3** PRAXIS, Kristiansand, Norway, **4** Department of Nursing and Health Science, University of Agder, Kristiansand, Norway

* rigmor.berg@fhi.no

**Data Availability Statement:** All relevant data are within the manuscript and its Supporting Information files.

**Funding:** This study did not have any funding. One author, Ms. Page, was employed with PRAXIS

## Abstract

### Background

The practice of involving people living with HIV in the development and provision of health-care has gained increasing traction. Peer-support for people living with HIV is assistance and encouragement by an individual considered equal, in taking an active role in self-management of their chronic health condition. The objective of this systematic review was to assess the effects of peer-support for people living with HIV.

### Methods

We conducted a systematic review in accordance with international guidelines. Following systematic searches of eight databases until May 2020, two reviewers performed independent screening of studies according to preset inclusion criteria. We conducted risk of bias assessments and meta-analyses of the available evidence in randomised controlled trials (RCTs). The certainty of the evidence for each primary outcome was evaluated with the Grading of Recommendations Assessment, Development, and Evaluation system.

### Results

After screening 219 full texts we included 20 RCTs comprising 7605 participants at baseline from nine different countries. The studies generally had low risk of bias. Main outcomes with high certainty of evidence showed modest, but superior retention in care (Risk Ratio [RR] 1.07; Confidence Interval [CI] 95% 1.02–1.12 at 12 months follow-up), antiretroviral therapy (ART) adherence (RR 1.06; CI 95% 1.01–1.10 at 3 months follow-up), and viral suppression (Odds Ratio up to 6.24; CI 95% 1.28–30.5 at 6 months follow-up) for peer-support participants. The results showed that the current state of evidence for most other main outcomes (ART initiation, CD4 cell count, quality of life, mental health) was promising, but too uncertain for firm conclusions.

towards the end of conducting this review. That is, the work on the review was initiated and most of the work completed before Ms. Page was employed with PRAXIS. Her work on this study was not part of her work as an employee of PRAXIS. PRAXIS had no role in this study: the organization did not play a role in the study design, data collection and analysis, decision to publish, preparation of the manuscript or financial support in the form of authors' salaries or research materials.

**Competing interests:** The authors do not have any conflict of interest. For a few months while we were conducting the study, Ms. Page was an employee of PRAXIS, Kristiansand. This does not alter our adherence to PLOS ONE policies on sharing data and materials.

## Conclusions

Overall, peer-support with routine medical care is superior to routine clinic follow-up in improving outcomes for people living with HIV. It is a feasible and effective approach for linking and retaining people living with HIV to HIV care, which can help shoulder existing services.

## Trial registration

CRD42020173433.

## Introduction

With 33 million lives lost so far and 38 million people living with HIV at the end of 2019, HIV remains a worldwide public health concern. Although global and national efforts have reduced the HIV incidence overall, HIV infections are on the rise in some countries and regions, particularly among key populations [1]. In most geographic areas, key population groups that account for over 95% of new HIV infections are men who have sex with men (MSM), people who inject drugs (IDU), people in prisons, sex workers and their clients, and transgender people. In the WHO African region, however, where over two thirds of people living with HIV live, HIV is prevalent among the general population [1].

Due to improved access to effective HIV prevention, diagnosis, treatment and care, HIV has become a manageable chronic health condition for most people. Yet, close to one in five people living with HIV do not know their status, and at the end of 2019, one third were not receiving antiretroviral therapy (ART) [1]. The WHO recommends that all people living with HIV are offered ART, because it makes HIV a manageable chronic health condition, saves lives, and contributes to reducing HIV transmissions [1, 2]. Unfortunately, ART provisions in high-endemic settings are challenged due to shortages linked to universal healthcare coverage [3]. Furthermore, key populations tend to have less access to ordinary healthcare services in all settings, and consequently also ART [4], and they face social and legal barriers to accessing treatment [5]. For example, stigma, which forty years into the epidemic continues to negatively affect the health and well-being of people living with HIV [6], has been found to play an important role in accessing treatment when HIV services are not integrated with other healthcare services [5, 7]. Even when people living with HIV can be reached and have access to care, they are often not taking the medications provided and fall out of care [8]. With this confluence of factors, an increasing burden for people living with HIV is coinfections and other comorbidities, with non-communicable diseases and mental health disorders as some of the most prevalent comorbidities [5, 9]. It has been estimated that about half of all people living with HIV meet criteria for one or more mental health disorder [10]. Research suggests that mental health disorders are not only associated with suboptimal HIV treatment outcomes, such as late ART initiation, poor ART adherence, and lack of viral suppression, but also delayed HIV diagnosis [11].

The effectiveness of a range of interventions designed to improve retention in care, ART initiation and adherence, stigma, and mental health of people living with HIV has been reviewed [12–18]. Interventions involving peer-support are both highlighted as a promising approach and appears to be an established strategy in many settings. Recently, the National Association of People With HIV Australia published the Australian HIV Peer Support

Standards. The standard aims to ensure that peer-support is provided to people living with HIV by people living with HIV, and that the peer-support is tailored to the needs of specific populations [19]. A similar standard is well-established in the U.K. [20]. Peer-support among people living with HIV has a long history. Already in the 1980s, groups of people living with HIV were supporting each other, sharing knowledge, and advocating for better treatment and care [20]. Today, there are diverse terms for peer-support interventions and somewhat different conceptualizations [21–23], but since the introduction of ART, peer-support has become a more tailored, person-centered outreach to provide linkage to and adherence to HIV care, as well as support people living with HIV in taking an active role in self-management of their chronic health condition [5, 20]. Positively UK specifies that peer-support is a relationship in which people are equal partners and the focus is on mutual learning and growth [20]. Dennis et al. similarly defines peer-support as "the giving of assistance and encouragement by an individual considered equal" ([24] p. 323). These conceptualizations of peer-support are in alignment with Fisher et al. [25] and The Peers for Progress program [23], who identify four key functions for peer-support: assistance in daily management, social and emotional support, linkage to clinical care and community resources, and ongoing support related to chronic disease. A peer is thus someone who shares common characteristics (e.g., age, sex, disease status) with the supported individual, such that the peer can relate to and empathize with the individual on a level that a non-peer would be unable to [26]. As such, peer-support fits within a social support model [27]. Within this model, peer-support has the potential to reduce feelings of isolation and loneliness, provide information, and promote behaviors that improve personal health, well-being, and health practices [27].

While there is increased recognition that peer-support, as a complement to general healthcare services, contributes to meeting the healthcare needs of people living with HIV [5, 18, 20, 25], several systematic reviews on people at risk for or living with HIV report that there is limited and mixed evidence on peer-support [12, 14, 17, 22, 28–30] and a systematic assessment of the global best-available evidence on the effectiveness of personalized peer-support for people living with HIV has not yet been undertaken. However, two reviews are thoroughly informative, whilst including a variety of study designs, literature searches ending in 2014, and lacking meta-analyses. First, Simoni and colleagues [22] cast a wide net with few delimiters and reviewed 117 papers published until 2010 on peer-interventions for people living with HIV. The authors did not assess the studies' risk of bias. By summing the number of studies with a supportive result, they found support for peer-based interventions, except for outcomes assessed in terms of biomarkers and other variables not self-reported, and most of the positive results were found in studies lacking randomization and relevant control groups. Similarly, Genberg and colleagues [29] investigated the effectiveness of using HIV-positive peers to bolster linkage, retention, and/or adherence to ART. They included any study with evaluation findings of a peer-based intervention (peer was defined as HIV-positive) for people living with HIV, and concluded that the findings from nine studies were mixed. Additionally, two other recent systematic reviews investigated the effectiveness of interventions related to peer-support. The most recent, on the effects of peer-led self-management interventions on ART adherence and patient-reported outcomes, found that the findings from thirteen controlled studies, conducted primarily in the U.S., showed unclear but promising effects [17]. Lastly, the effectiveness of the use of peers for achieving ART adherence and viral suppression was assessed in a systematic review with network meta-analyses [18]. Kanters et al. [18] found few studies on viral suppression and were unable to conclude, but the authors determined that using peer supporters plus telephone was superior to standard care in improving ART adherence, both in low and middle income settings and elsewhere.

The objective of the present systematic review was to examine the effects of peer-support for people living with HIV, such that decision makers have solid evidence to implement interventions that improve the management of healthcare and lives of people living with HIV. Our research question was: what is the effectiveness of peer-support on medical and psychosocial outcomes for people living with HIV?

## Materials and methods

We conducted a systematic review in accordance with guidelines set forth in the Cochrane Handbook for Systematic Reviews of Interventions [31]. The pre-specified protocol was registered in PROSPERO (S1 File) and we report in accordance with the PRISMA checklist [32] (S1 Table).

### Eligibility criteria

For the search and screening processes, we applied the (S)PICO model, which directs attention to the study design, population, intervention, comparison, and outcomes [33]. Eligible study designs were randomised controlled trials (RCT) and non-RCTs. We pre-specified that if a solid number of RCTs were included, we would not include non-RCTs. Study participants included people living with HIV, 18 years and older. Interventions had to focus on peer-support interventions or programmes [24]. To be included, peer-support had to be given to a person living with HIV by another person living with HIV for a minimum of 60 minutes face-to-face interaction because an underlying premise of peer-support is personalized interaction. Both those receiving and those providing peer-support needed to be aged 18 years or older. All comparison conditions were eligible. Studies had to provide data for at least one of the following primary outcomes (i.e. considered to be the most important): retention in care, ART initiation, ART adherence, CD4 cell count, viral load (or viral suppression/failure), quality of life, and mental health. Secondary outcomes considered were adherence to care, HIV risk behaviors, and stigma. We included all settings, but only studies published in English or Scandinavian languages (Norwegian, Swedish, Danish) were included. We included studies published after 1981, because this is the first year there were any publications about HIV/AIDS. Unpublished reports, briefs, and preliminary reports were considered for inclusion on the same basis as published reports.

### Search strategy for the identification of studies

First, to identify relevant keywords and to search for existing systematic reviews, we conducted a preliminary search in PROSPERO and the JBI Database of Systematic Reviews and Implementation Reports. The main literature identification strategy consisted of searches in the following eight international electronic databases: MEDLINE (OVID), MEDLINE In-Process (OVID), EMBASE (OVID), CINAHL (EBSCOhost), PsycINFO (OVID), SocINDEX (EBSCOhost), Social Work Abstracts (EBSCOhost), and BASE (Bielefeld Academic Search Engine). To conduct the searches, we used a piloted strategy incorporating subject headings (e.g., MeSH in MEDLINE) and text words in the titles and abstracts, adapted for each database. One of the reviewers (AØR) conducted the searches together with an information search specialist. The search in MEDLINE is shown in S2 File. Additionally, we screened reference lists of the included studies and relevant literature reviews, conducted a search for grey literature in Scopus, Google Scholar, BASE, the UK government website, CORE, and searched for ongoing studies on clinicaltrials.gov and WHO Trial Register. These processes were repeated until no new references were identified. The searches were completed in May 2020.

## Study selection

We imported the search records to EndNote [34] and deleted duplicates. The search records were then imported to Rayyan QCRI [35] and the literature screening was carried out by two reviewers in a two-stage process, with increased scrutiny of the records based on the inclusion criteria of the review. At each level, we evaluated the identified records independently of one another using a pre-developed inclusion form. The final determination to include or exclude was made together. Throughout the process of screening titles/abstracts, and full-texts, differences were resolved through discussion and consensus. We were not blinded to the authors or other information when assessing the records. Only those studies meeting all inclusion criteria were included.

## Data extraction and risk of bias assessment

The data extraction was carried out by one reviewer, and a second reviewer checked the completeness and accuracy of the data extracted. A piloted data extraction form developed for the study was used, to ensure standardization in data extracted. We extracted data regarding: publication characteristics (type of publication, author, year), study characteristics (country, study design, sample size), characteristics of the study participants (e.g. gender), characteristics of the peer-support intervention and the control condition (e.g. duration, content, setting, theoretical basis), and study results (outcome data including follow-up). We obtained all data from published records and were not in contact with study investigators.

We planned to appraise the risk of bias of included controlled studies using design-specific checklists. Because we only included RCTs, we used the Cochrane Risk of Bias (RoB) tool [31]. RoB was done at outcome level, with the main biologically measured outcome when available. Two researchers conducted independent risk of bias assessments and then agreed on a final RoB evaluation.

## Data analysis and assessment of the certainty of the evidence

We extracted and present dichotomous and continuous data for all eligible outcomes when postscores for both intervention and control groups were reported by study authors, allowing for comparison. We extracted crude data and, when such data were available, adjusted outcome data (adjusted comparison (effect) estimates and their standard errors or 95% confidence intervals, CI). We present dichotomous outcomes as the number of events and number of people in groups as proportions, risk ratio (RR) or odds ratio (OR) as appropriate. We present continuous outcomes as mean difference (MD) and standard deviations (SD), or use the most appropriate presentation based on the available data in the included studies.

We sorted the included studies according to comparisons and outcomes, and evaluated the characteristics of the studies' PICO. When they were considered sufficiently similar, and data were available, we conducted meta-analyses. We based judgments about whether meta-analyses were appropriate on recommendations in the Cochrane Handbook [31]. We used Mantel-Haenszel random effects meta-analysis for dichotomous outcomes, and inverse variance random effects meta-analysis for continuous outcomes. For dichotomous outcomes we presented the relative risk and the corresponding 95% CI. For continuous outcomes we analysed the data using (standardized) mean difference ((S)MD) with the corresponding 95% CI. We also examined between-study heterogeneity using visual inspection of CIs, the Chi-square test, and I-square statistic, quantifying the degree of heterogeneity as follows [31]: 0% to 40% might not be important; 30% to 60% may represent moderate heterogeneity; 50% to 90% may represent substantial heterogeneity; 75% to 100% considerable heterogeneity. When possible, we explored reasons for heterogeneity. We used RevMan version 5.4, the latest version of the

Cochrane meta-analysis software [36]. When the studies' PICOs or results were considered too heterogeneous to pool statistically, or data were unavailable, we reported the results narratively, in text and tables.

We assessed the certainty of the evidence for each primary outcome using GRADE (Grading of Recommendations Assessment, Development, and Evaluation). One reviewer performed the assessment, and another checked the data and assessment. The grading represents our certainty in the evidence of effect based on the available studies. The GRADE approach has five criteria for possible downgrading of the certainty in the evidence: study limitations, inconsistency between studies, indirectness of evidence, imprecision, and reporting bias. We provide justification for decisions to down-grade the ratings using footnotes and comments. We used the four standard definitions in grading the certainty of the evidence: high, moderate, low, very low. For more information about the GRADE system, see gradeworkinggroup.org and publications by the GRADE Working Group (e.g. [37]).

## Results

As shown in Fig 1, of 5470 unique records, we screened 219 full-texts and included 20 RCTs [38–57]. Most excluded studies were excluded for two or more reasons, but the most common primary reasons for exclusion were that the intervention was not peer-support as per our inclusion criteria, providers were not individuals living with HIV, and the study design was not RCT. Given the differences in inclusion criteria–particularly concerning population and the peer intervention–and publication years, there was minimal overlap in included studies between our review and previously published, related reviews [17, 18, 22, 29].

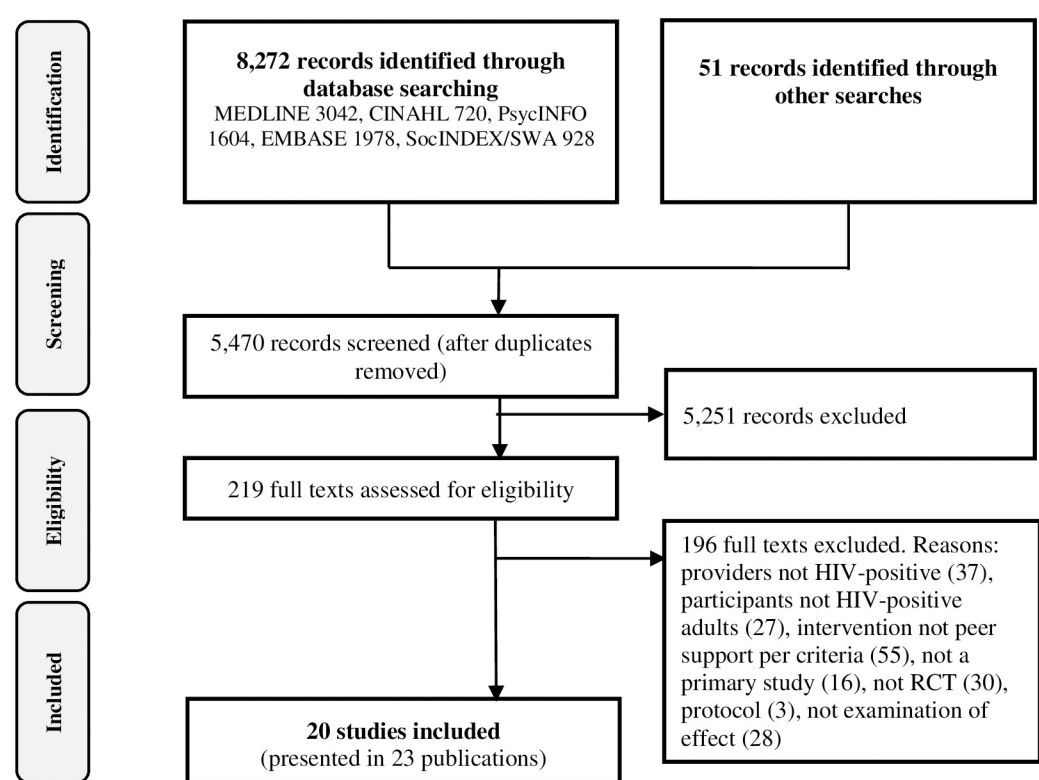

**Fig 1. PRISMA flow diagram of literature reviewing process.**

## Characteristics of the included studies

The included studies were published between 2001 and 2020 and three were cluster RCTs (Table 1).

For ease of reporting, studies are denoted *k*. The studies were conducted in nine different countries, with half being from USA (k = 10). Other studies were from Uganda (k = 2), Kenya (k = 2), and there was one study each from China, Mozambique, Nigeria, Spain, South Africa, and Vietnam. The 20 studies included 7605 participants at baseline (range 20–1336) and most included both males and females, but three targeted MSM only, one trial was tailored for women only, and three studies included also transgender people. The populations were hetero-geneous, as three studies included people who inject drugs, six studies had either people who were newly diagnosed with HIV or treatment naïve, three studies had people who were on ART, and two studies were aimed at people who were non-adherent. While all interventions were usual medical care plus peer-support, they were tailored to country or community con-text and their scope, intensity, and theoretical foundations differed considerably. The number of sessions ranged from 1–36 (median 7) up to 26 months and eleven were based on one or more theories or frameworks, such as the Social cognitive theory (k = 3), the Information-Motivation-Behavioral skills model (k = 3), or the Wellness motivation theory (k = 2). The most common aspects, or key functions of peer-support [23, 25], were linkage to clinical care and community resources, often combined with assistance in daily management and social and emotional support. In all except four of the trials, the comparison group received usual care, which typically encompassed regular clinic visits for medical care, case management, and support services. The four time-equivalent active comparisons were counselling (healthy eat-ing and exercise), didactic education, psychoeducation, and video discussions. The most com-monly reported outcomes were ART adherence (k = 8), viral suppression (k = 8), risk behaviors (k = 7), CD4 cell count (k = 5), and retention in care (k = 5).

## Risk of bias

Fig 2 shows that the studies had generally low risk of bias, particularly concerning selection bias, reporting bias, and other bias. Six studies had high risk of attrition bias and one study had also high risk of performance bias and of detection bias (S2 Table).

## Study results

In our protocol we specified seven primary outcomes and three secondary outcomes, all of which were reported by at least one of our included studies, except adherence to care. How-ever, the outcomes were often measured differently (biologically, self-report, different instru-ments, visual analogue scale, etc.), operationalized with different cut-off scores, examined at different follow-up times (from 3 to 24 months), reported with different effect measures, and sometimes reported without the necessary data to pool in meta-analyses (e.g., SD was not reported). As a result, while we found that most studies' PICOs were conceptually sufficiently similar to meta-analyse, we could at most pool six studies. In total, we could statistically pool seven outcomes. The low number of studies with similar outcomes prevented sensitivity analy-ses and an assessment of dissemination bias using the funnel-plot technique.

We report the effect of peer-support for each of the primary and secondary outcomes below. The meta-analytic results are presented in Figs 3–9. Outcome results that could not be meta-analysed are shown in Table 2. Our GRADE assessments, described in Table 3, show that our certainty in the effect estimates ranged from very low to high.

**Primary outcomes.** *Retention in care*. Five studies, four from USA and one from Uganda, reported on retention in care. More peer-support participants than control

**Table 1. Characteristics of the included studies (studies listed in alphabetical order: K = 20).**

| Study (country) Study design | Population | Intervention | Comparison | Outcomes |
|---|---|---|---|---|
| **Brashers et al., 2017 [38] (USA) RCT** | N = 98, m&f, newly diagnosed | "Living with HIV/AIDS: Taking Control". Based on Uncertainty management theory. 6 sessions (over 1.5 mo). | Usual care | Social support |
| | | | | Depressive symptoms |
| **Broadhead et al., 2012 [39] (USA) RCT** | N = 78, m&f, IDUs | "CHAMPS". Based on principles of harm reduction and theory of group-mediated social control. 12 sessions (over 3 mo). | Usual care | Risk behaviors |
| | | | | Social functioning |
| **Cabral et al., 2018 [40] (USA) RCT** | N = 348, m&f, ethnic minorities | Based on the social support framework. 7 sessions (over 6 mo). | Usual care | Retention in care |
| | | | | Viral suppression |
| | | | | Quality of life |
| **Chang et al., 2010 [41] (Uganda) cRCT** | N = 1336, m&f | Support, counseling and education. ~13 sessions (over 26 mo). | Usual care | ART adherence |
| | | | | ART initiation |
| **Chang et al., 2015 [42] (Uganda) RCT** | N = 442, m&f, treatment-naïve | Based on a situated information, motivation and behavioral skills framework. 12 sessions (over 12 mo). | Usual care | ART initiation |
| | | | | Virologic failure |
| | | | | Retention in care |
| | | | | Risk behaviors |
| **Coker et al., 2015 [43] (Nigeria) RCT** | N = 600, m&f, treatment-naïve | Support, counseling and education. N$^O$ sessions not stated (over 9 mo). | Usual care | Viral suppression |
| **Cunningham et al., 2018 [44] (USA) RCT** | N = 356, m&f&tg, MSM, IDUs newly released from jail | "LINK LA Peer Navigation": Based on patient navigation and Social cognitive theory. 12 sessions (over 6 mo). | Usual care | Viral suppression |
| | | | | Retention in care |
| | | | | ART adherence |
| | | | | Risk behaviors |
| Cuong et al., 2016 [45] (Vietnam) cRCT | N = 640, m&f, treatment-naïve | Support, counseling and education. Weekly home visits. N$^O$ sessions and time period not stated. | Usual care | Virologic failure |
| | | | | CD4 cell count |
| | | | | Quality of life |
| | | | | Stigma |
| **Enriquez et al., 2015 [46] (USA) RCT** | N = 20, m&f, non-adherent to ART | "Ready": based on Wellness motivation theory and Social cognitive theory. 7 sessions (over 1.5 mo). | Healthy eating and exercise counselling | Viral suppression |
| **Enriquez et al., 2019 [47] (USA) RCT** | N = 30, m&f&tg, non-adherent to ART | "Peers Keep It Real": based on "Ready" which is based on Wellness motivation theory and Social cognitive theory. 7 sessions (over 1.5 mo). | Usual care | Viral suppression |
| **Fogarty et al., 2001 [48] (USA) RCT** | N = 322, f | Support, counseling and education–focus on birth control use. N$^O$ sessions not stated (over 6 mo). | Usual care | Risk behaviors |
| **Giordano et al., 2016 [49] (USA) RCT** | N = 460, m&f, newly diagnosed or out-of-care | "Mentor Approach for Promoting Patient Self-care": focus on managing HIV. 7 sessions (over 2.5 mo). | Didactic education about safer sex and safer drug use | Retention in care |
| | | | | Viral suppression |
| | | | | ART adherence |
| | | | | CD4 cell count |
| | | | | Quality of life |
| **Graham et al., 2020 [50] (Kenya) RCT** | N = 60, MSM | "Shikamana": based on Next step counseling/ Motivational interviewing. ~4 sessions (over 6 mo). | Usual care | Viral suppression |
| | | | | ART adherence |
| Liu et al., 2018 [51] (China) RCT | N = 367, MSM, newly diagnosed | Based on adapted Information-Motivation-Behavioral skills model. 1 60-minute session. | Usual care | Risk behaviors |
| **McKirnan et al., 2010 [52] (USA) RCT** | N = 313, MSM | "Treatment Advocacy Program": focus on transmission risk reduction and coping. 6 sessions (over 12 mo). | Usual care | Risk behaviors |

(*Continued*)

**Table 1.** (Continued)

| Study (country) Study design | Population | Intervention | Comparison | Outcomes |
|---|---|---|---|---|
| Pearson et al., 2007 [53] (Mozambique) RCT | N = 350, m&f, treatment-naïve | Support, counseling and education with modified directly observed therapy. 30 sessions (over 2.5 mo). | Usual care | ART adherence |
| | | | | CD4 cell count |
| **Purcell et al., 2007 [54] (USA) RCT** | N = 966, m&f&tg, IDUs | "INSPIRE": based on empowerment theory, peer leadership, Social learning theory, Social identity theory, Information-Motivation-Behavioral skills model. 10 sessions (over 2.5 mo). | Video discussions (8 sessions) | Retention in care |
| | | | | ART adherence |
| | | | | Risk behaviors |
| Ruiz et al., 2010 [55] (Spain) RCT | N = 240, m&f, on ART | Support, counseling and education. 4 sessions (over 6 mo). | Psychoeducation given by health professional | ART adherence |
| | | | | Viral suppression |
| | | | | Psychological distress |
| **Selke et al., 2010 [56] (Kenya) cRCT** | N = 239, m&f, on ART | Support, counseling and education. Delivered medications. 12 sessions (over 12 mo). | Usual care | CD4 cell count |
| | | | | ART adherence |
| | | | | Virologic failure |
| Wouters et al., 2014 [57] (South Africa) RCT | N = 340, m&f, on ART | Based on the family functioning framework: focus on family dynamics. 36 sessions (over 18 mo). | Usual care | CD4 cell count |
| | | | | Mental health |
| | | | | Stigma |

Legend: ART = antiretroviral therapy, cRCT = cluster randomised controlled trial, f = female, IDU = injection drug user, m = male, mo = month, MSM = men who have sex with men, N$^{O}$ = number, tg = transgender.

participants were retained in care at both 6-months follow-up and 12-months follow-up (Fig 3. RR = 1.05, 95% CI = 0.92, 1.20; RR = 1.07, 95% CI = 1.02, 1.12, respectively). While we have high certainty in the 12-months follow-up result, the meta-analysis for retention in care at 6-months follow-up have high statistical heterogeneity (Chi$^2$ = 6.88, p = 0.03; I$^2$ = 71%) and we have moderate certainty in the estimate. Due to the low number of studies we could not conduct subgroup analyses or meta-regression, and we found no explanations for the statistical heterogeneity linked to clinical- or methodological diversity. Thus, it was unclear why there is statistical heterogeneity, beyond that this meta-analysis has one outlying study result, specifically the large positive effect in Cunningham et al. [44] compared to Giordano et al. [49] and Purcell et al. [54]. The result must be interpreted with caution. Another study, Cabral et al. [40], could not be included in the pooled analyses, but found that time to 4-month gap in HIV care favoured peer-support (HR = 0.82, 95% CI = 0.72, 0.94).

*ART initiation*. Two studies, both by the same research group in Uganda, reported on ART initiation, finding no statistically significant differences between the groups at any follow-up time (months 6, 12, 18, 24) (Table 2 and Fig 4).

*ART adherence*. Eight RCTs–from Kenya, Mozambique, Spain, Uganda, USA–reported on ART adherence from 3–12 months follow-up. The studies that could be pooled generally defined adherence as having taken 90% of antiretroviral medications in the prior week, and found equivalent estimates across time, favouring peer-support (Fig 5). The meta-analytic result at 12-months follow-up had high heterogeneity (Chi$^2$ = 9.47, p = 0.02; I$^2$ = 68%). Again, there is an absence of a clear explanation for the statistical heterogeneity linked to clinical- or methodological diversity, we can only observe that this meta-analysis appears to have one outlying study result, specifically the negative effect in the study by Selke et al. [56]. The result must therefore be interpreted with caution.

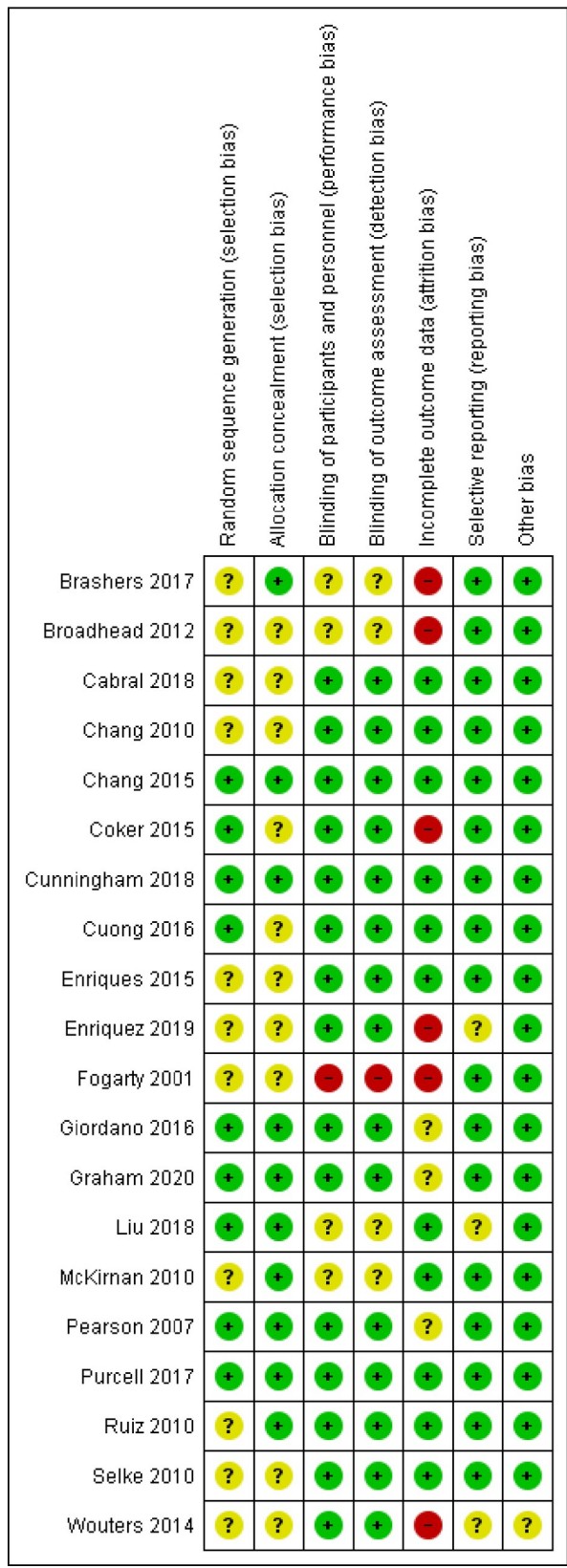

**Fig 2. Risk of bias in included studies.**

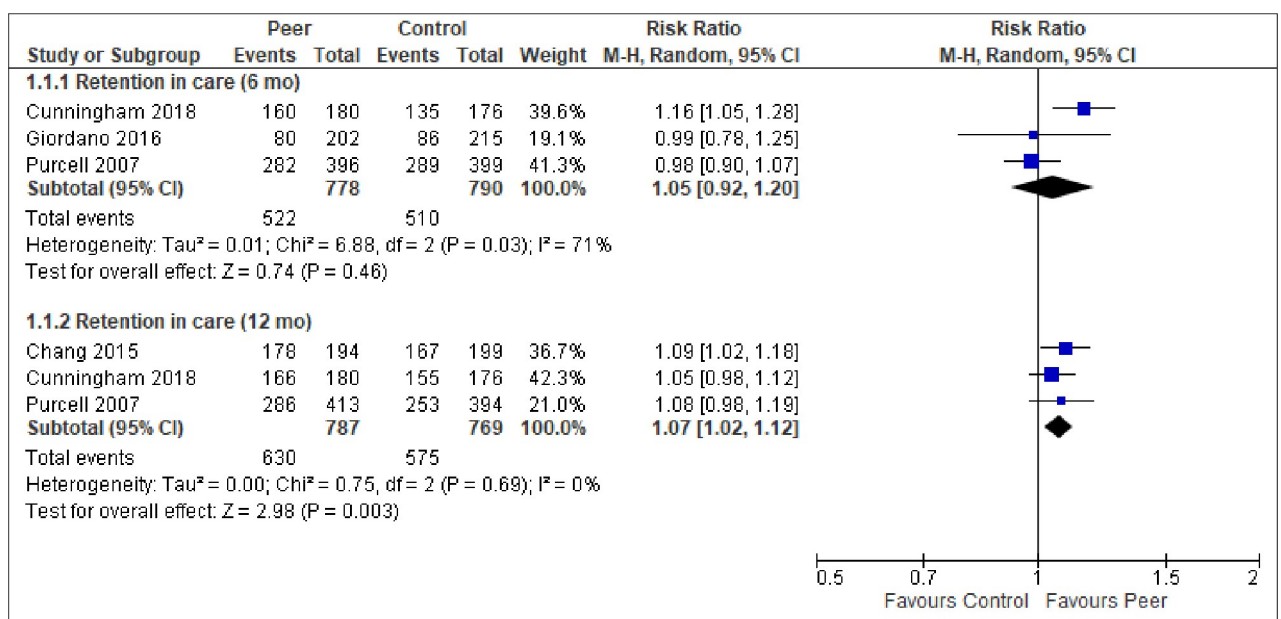

**Fig 3. Meta-analyses of outcome retention in care.**

*CD4 cell count.* While five studies–conducted in Kenya, South Africa, USA and Vietnam–reported on CD4 cell count, the outcomes were reported too differently to be statistically pooled. As shown in Table 2, none of the studies found a statistically significant difference in CD4 cell count between the groups, across different follow-up times (6–24 months).

*Viral outcomes (HIV RNA).* Eight of the included studies reported on viral suppression, using slightly different cut-offs, but generally about 100 copies/mL. The two studies reporting viral suppression at 3-months and 12-months follow-up had too heterogeneous results to be statistically pooled (Table 2). Similarly, it was statistically unwarranted to pool all the seven studies reporting viral suppression at 6-months. While the meta-analytic result–based on studies conducted in Spain, Nigeria, and USA–found no statistically significant difference between the groups (Fig 6), the three studies that could not be pooled all found a statistically significant effect at 6-months follow-up, favouring peer-support (Table 2). We note that the results were heterogeneous. Similarly, four studies, carried out in Kenya, Uganda, and Vietnam, reported on virologic failure at months 6, 12, 18 and 24. Virologic failure was defined as >400 to >1000 copies/ml or specified in even greater detail as primary virologic failure if VL >1000 copies/ml

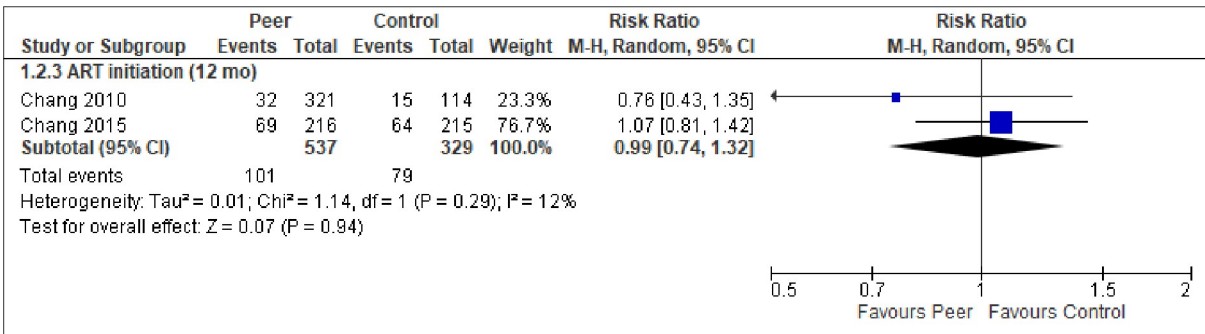

**Fig 4. Meta-analysis of outcome ART initiation.**

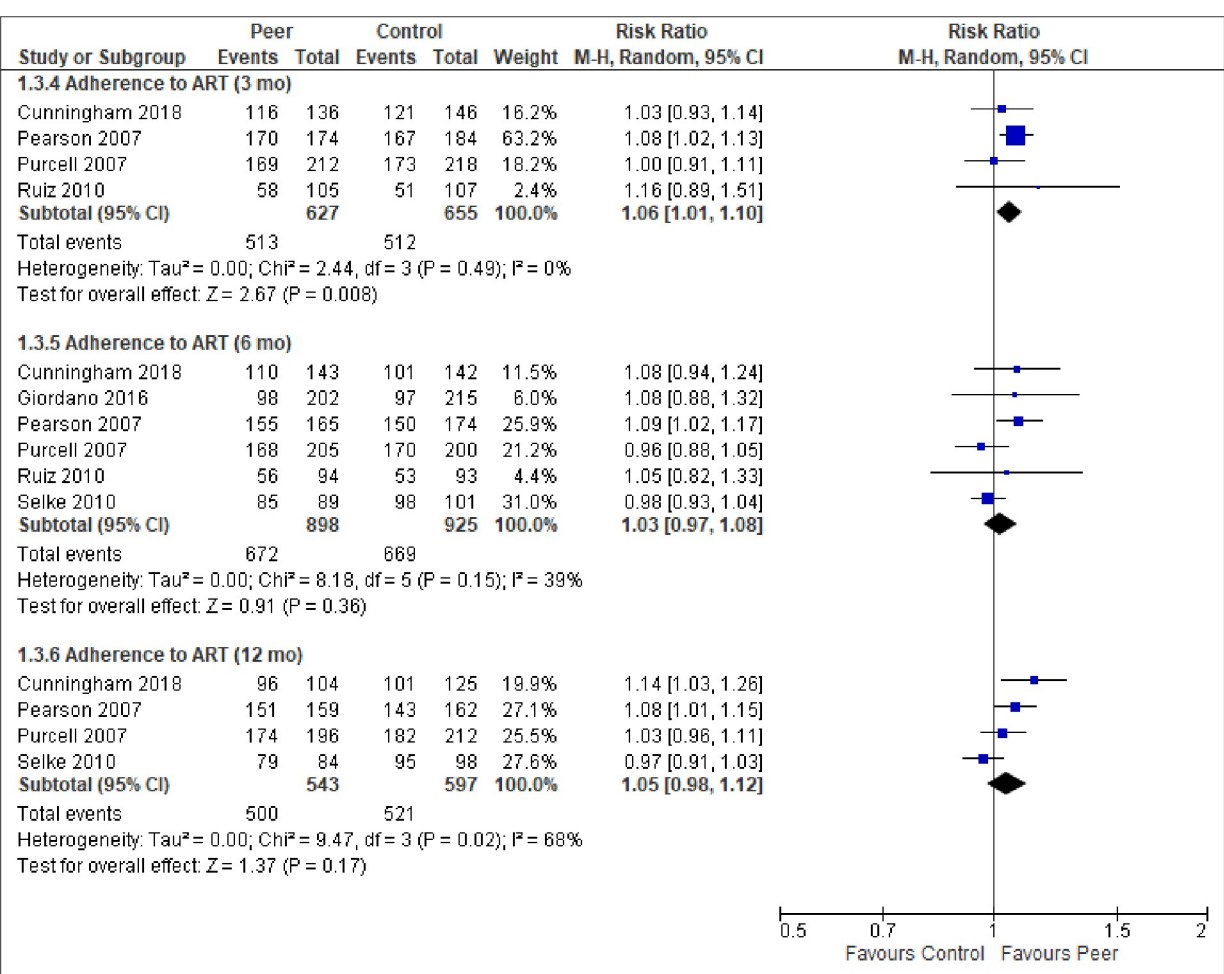

**Fig 5. Meta-analyses of outcome adherence to ART.**

after 6 months of ART initiation, or, secondary virologic failure if VL was undetectable (<200 copies/ml) after 6 months of ART initiation and then became >1000 copies/ml at any time point during the follow-up. Despite these operational nuances, statistically, there was low

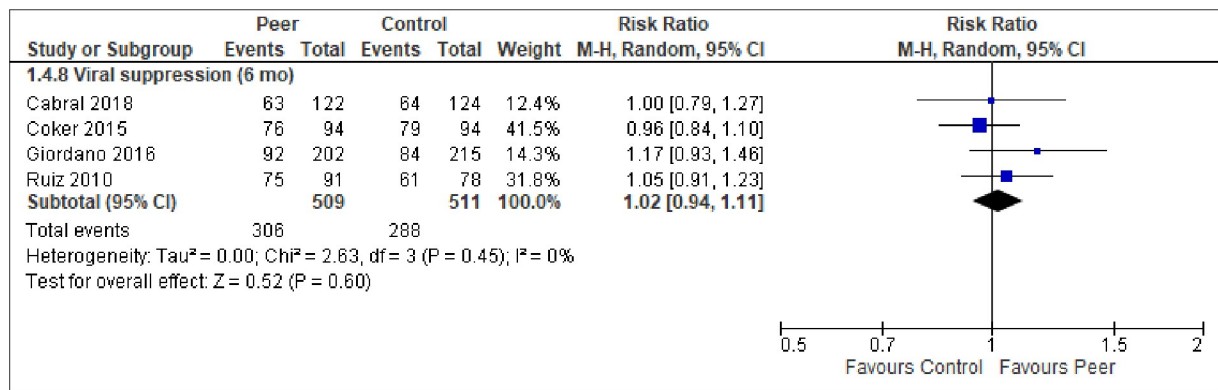

**Fig 6. Meta-analysis of outcome viral suppression.**

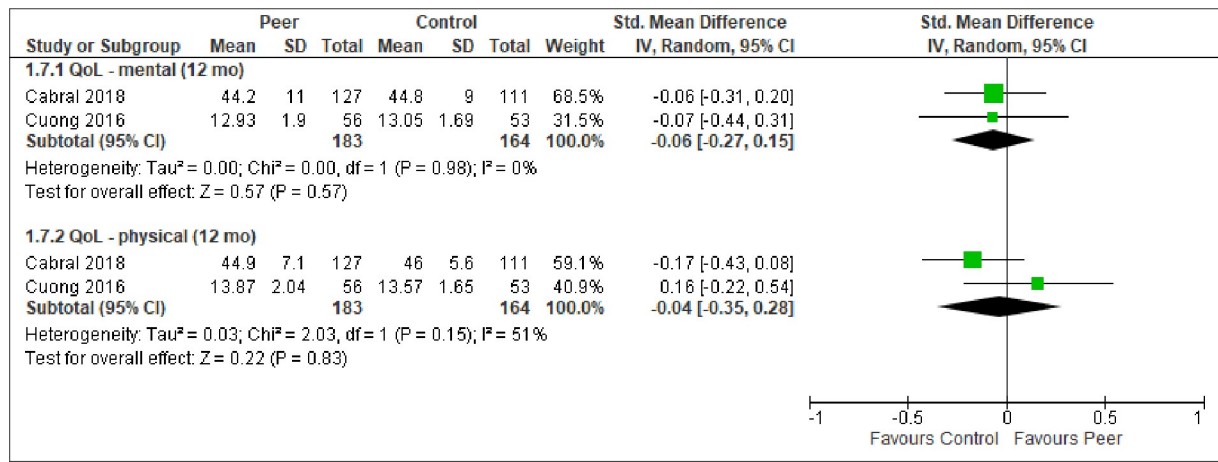

**Fig 7. Meta-analyses of outcome virologic failure.**

**Fig 8. Meta-analyses of outcome quality of life.**

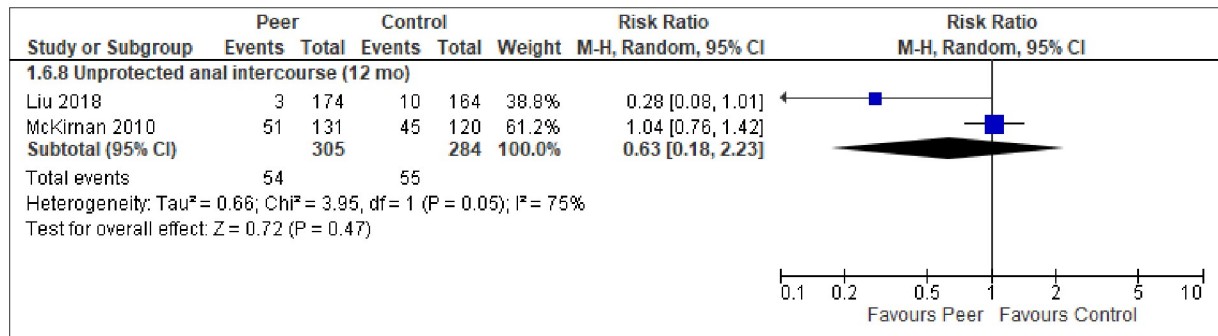

**Fig 9. Meta-analysis of outcome unprotected anal intercourse.**

inconsistency between the studies and the results generally favoured peer-support (Fig 7). Of the four meta-analyses of virologic failure, only the meta-analysis for virologic failure at 24-months follow-up was statistically significant, in favour of peer-support (RR = 0.54, 96% CI = 0.31, 0.94, moderate certainty).

*Quality of life.* Only three studies measured and reported on quality of life, and did so quite differently, using SF-8, SF-36, and WHOQOL-HIVBREF. Two studies were from USA and one was from Vietnam. By standardizing, we could pool the results of two studies on two different domains of quality of life: mental- and physical quality of life (Fig 8). Neither the meta-analyses nor the results in the third study [48] found a statistically significant difference between the groups with respect to quality of life (Table 2).

*Mental health.* Only two studies, one from Spain and one from USA, reported on mental health (Table 2). They used different scales for depression, but while both studies found that participants in the peer-support group fared better, neither found a statistically significant difference between the two groups. Other related outcomes were social support (one study) and social functioning (two studies), but the studies detected no statistically significant differences between peer-support and control regarding these outcomes (Table 2).

**Secondary outcomes.** Of the outcomes adherence to care (no studies), HIV risk behaviors, and stigma we could only conduct a meta-analysis of unprotected anal intercourse (an HIV risk behavior). It showed considerable heterogeneity across studies and no statistically significant difference between the groups (Fig 9). As seen in Table 2, three studies reported similar sex-related HIV risk behaviors, such as multiple sexual partners and use of condoms, but none found any statistically significant differences between peer-support and control. With regard to drug-related HIV risk behaviors, three studies reported on use of hard- or illicit substances or needle sharing. One study, with newly diagnosed MSM in China, found a decreased odds of illicit drug use for peer-support participants at 12 months, while the other two studies found no statistically significant differences between the groups. Finally, two studies examined HIV stigma among the participants. The study from Vietnam found an identical result in the two groups. The other, a study from South Africa, which had unclear or high risk of bias in five domains [57], reported that receiving peer-support significantly increased the participants' level of felt stigma.

## Discussion

To our knowledge, while there are several reviews of community-based interventions for people living with HIV [4, 17, 18, 22, 28–30, 58], this is the first systematic review and meta-analysis to assess the effects of personalized, face-to-face peer-support on a variety of

**Table 2. Study outcomes and effect estimates.**

| Study | Outcome (follow-up) | Result/ Effect estimate (95%CI) |
|---|---|---|
| Retention in care | | |
| Cabral et al. 2018 [40] | Time to 4-mo gap in HIV care | HR = 0.82 (0.72–0.94) |
| ART initiation | | |
| Chang et al. 2010 [41] | ART initiation (6 mo) | RR = 0.93 (0.63–1.37), p = 0.71 |
| | ART initiation (18 mo) | RR = 0.79 (0.22–2.31), p = 0.71 |
| | ART initiation (24 mo) | RR = 0.31 (0.06–1.65), p = 0.17 |
| ART adherence/pill count adherence | | |
| Chang et al. 2010 [41] | Pill count adherence <95% (12 mo) | RR = 0.57 (0.23–1.37), p = 0.21 |
| | Pill count adherence <100% (12 mo) | RR = 1.09 (0.87–1.37), p = 0.44 |
| | Any missed doses (12 mo) | RR = 0.91 (0.71–1.19), p = 0.50 |
| Graham et al. 2020 [50] | Visual analogue scale adherence ≥80% (6 mo) | OR = 1.53 (0.63–3.75), p = 0.35 |
| CD4 cell count | | |
| Cuong et al. 2016 [45] | Increase in median CD4 count from baseline to month 24 | diff = 13 cells/µl, p = 0.77 |
| Giordano et al. 2016 [49] | CD4 ≥350 cells/µL (6 mo) | RR = 1.11 (0.89–1.39) |
| | CD4 ≥500 cells/µL (6 mo) | RR = 1.20 (0.94–1.53) |
| Pearson et al. 2007 [53] | Mean CD4 cell count (6 mo) | 140.6 (12.5) vs 144.4 (12.0), ns |
| | Mean CD4 cell count (12 mo) | 176.4 (14.3) vs 176.0 (13.1), ns |
| Selke et al. 2010 [55] | Mean CD4 cell count (IQR) (6 mo) | 354 (232–451) vs 306 (214–410), p = 0.24 |
| | Mean CD4 cell count (IQR) (12 mo) | 404 (265–527) vs 358 (240–522), p = 0.50 |
| Wouters et al. 2014 [57] | Mean CD4 cell count (12 mo) | ns |
| Viral suppression | | |
| Cabral et al. 2018 [40] | Viral suppression (12 mo) | RR = 0.80 (0.64–0.99) |
| Cunningham et al. 2018 [44] | Viral suppression (3 mo) | RR = 1.31 (1.03–1.67) |
| | Viral suppression (12 mo) | RR = 1.38 (1.03, 1.85) |
| Enriquez et al. 2015 [46] | There was a stat.sig difference in viral load suppression/ medication adherence between groups, favoring peer support, p<0.01 (6 mo) | |
| Enriquez et al. 2019 [47] | The intervention increased the chance of viral load suppression by 5.2-fold (6 mo) | |
| Graham et al. 2020 [50] | Plasma viral load ≤40 copies /mL (6 mo) | OR = 6.24 (1.28–30.5), p = 0.02 |
| Ruiz et al. 2010 [55] | Viral suppression (3 mo) | RR = 0.98 (0.85, 1.14) |
| Quality of life | | |
| Giordano et al. 2016 [49] | Health-related quality of life, mean change from baseline (6 mo) | General health = 5.9 vs 7.96, p = 0.49 |
| | | Social function = 9.52 vs 4.73, p = 0.32 |
| | | Physical function = 6.06 vs 0.86, p = 0.19 |
| | | Physical limitation = 13.27 vs 4.14, p = 0.05 |
| Depressive symptoms | | |
| Brashers et al. 2017 [38] | Depression (Center for epidemiologic studies depression scale) (12 mo) | 17.43 (13.12) vs 22.43 (12.33), ns |
| Ruiz et al. 2010 [55] | Psychological distress (General Health Questionnaire) (6 mo) | 26% vs 28.3%, ns |
| Social support | | |
| Brashers et al. 2017 [38] | Social support satisfaction (6 items, self-designed) (12 mo) | 4.44 (1.70) vs 3.54 (1.66), ns |
| Social functioning | | |
| Broadhead et al. 2012 [39] | "The results show an increase in social functioning over time for the PDI group and almost no change in social functioning over time for the UCI group" (12 mo) | |
| Cunningham et al. 2018 [44] | SF-12 mental health (3 mo) | diff = 1.1 (-1.6 to 3.8), p = 0.42 |
| | SF-12 mental health (6 mo) | diff = -0.3 (-3.1 to 2.5), p = 0.84 |
| | SF-12 mental health (12 mo) | diff = -1.2 (-4.1 to 1.8), p = 0.44 |
| Sex-related risk behavior | | |
| Chang et al. 2015 [42] | Multiple sexual partners: 21 of 63 (33.3%) vs 18 of 53 (34.0%), PRR = 0.98 (0.61–1.60) (12 mo) | |

*(Continued)*

**Table 2.** (Continued)

| Study | Outcome (follow-up) | Result/ Effect estimate (95%CI) |
|---|---|---|
| Fogarty et al. 2001 [48] | Use of condom with main partner: "At the first transition, women in the enhanced group had 2.8 times the odds of progressing and less than half the odds of relapsing in their use of condoms with their main partner than did women in the standard group. This trend continued throughout the study, although behavior changes were not statistically different between the groups at the second and third transitions." | |
| Purcell et al. 2007 [54] | Unprotected vaginal or anal sex with HIV-negative/unknown serostatus partner (3 mo) | aOR = 1.22 (0.79–1.89) |
| | Unprotected vaginal or anal sex with HIV-negative/unknown serostatus partner (6 mo) | aOR = 1.32 (0.83–12.12) |
| | Unprotected vaginal or anal sex with HIV-negative/unknown serostatus partner (12 mo) | aOR = 1.01 (0.63–1.61) |
| Drug-related risk behavior | | |
| Cunningham et al. 2018 [44] | All hard substance use (3 mo) | diff = -0.07 (-0.21, 0.07), p = 0.33 |
| | All hard substance use (6 mo) | diff = -0.05 (-0.22, 0.12), p = 0.57 |
| | All hard substance use (12 mo) | diff = -0.09 (-0.25, 0.08), p = 0.31 |
| Liu et al. 2018 [51] | Illicit drug use (12 mo) | aOR = 0.32 (0.16–0.64) |
| Purcell et al. 2007 [54] | Lent a needle to or shared drug paraphernalia with HIV-negative /Unknown serostatus partner (3 mo) | aOR = 0.78 (0.49–1.21) |
| | As above (6 mo) | aOR = 0.68 (0.40–1.13) |
| | As above (12 mo) | aOR = 0.77 (0.42–1.41) |
| Stigma | | |
| Cuong et al. 2016 [45] | Internal AIDS-related stigma: 3.27 (SD 1.8) both groups, ns | |
| Wouters et al. 2014 [57] | Receiving peer adherence support significantly increased the level of stigma experienced at the second follow up: β = 0.31, p = 0.001 | |

Legend: SD = standard deviation; aOR = adjusted Odds Ratio; diff = Difference; HR = Hazard Ratio; mo = months; ns = non-significant; PRR = prevalence rate ratio; RR = risk ratio.

outcomes for people living with HIV. Our systematic review of 20 RCTs from 2001–2020 with generally low risk of bias shows that, overall, peer-support with routine medical care is feasible and superior to routine clinic follow-up in improving outcomes for people living with HIV. While the clinical effects were modest, we found moderate to high certainty of evidence for greater long-term retention in care, improved ART adherence, reduced risk of virologic failure, and better viral suppression with peer-support. That is, our findings demonstrate that peer-support is an effective approach for linking and retaining people living with HIV to HIV care, and for improving ART adherence and consequently viral suppression as well as avoiding virologic failure. In regard to overall completeness and applicability of the evidence, all key populations were represented in our included trials, except sex workers and their clients, and half of the studies were from USA, leaving other locations greatly influenced by HIV, such as Sub-Saharan Africa and Asia, underrepresented. While about half of the outcomes had moderate-high certainty, and we observe that peer-support has clear benefits, with the current state of evidence firm conclusions on the effects on ART initiation, quality of life, social support, and mental health are difficult to reach. However, we note that while the certainty of evidence for mental health was uncertain, the results are promising in that both studies assessing depressive symptoms found that participants in the peer-support group fared better [38, 55]. Importantly, across the trials, almost all outcomes favored peer-support, which rejects any potential suspicions of unfavorable effects from peer-support. The result regarding a negative effect on stigma [57] may be related to low degree of compatibility between the intervention and the context in which it was implemented. The researchers [57] conducted moderator analyses with family functioning, finding that living in vulnerable and dysfunctional families was a key factor explaining the negative impact of the intervention.

**Table 3. Certainty of evidence of effect of peer-support for people living with HIV.**

**Population:** People living with HIV
**Countries:** China, Kenya, Mozambique, Nigeria, Spain, Uganda, USA, Vietnam
**Intervention:** Peer-support
**Comparison:** Usual care or education/counselling

| Outcome, follow-up time | Anticipated absolute effects* (95% CI) | | Relative effect (95% CI) | No. of participants (Studies) | Quality of evidence (GRADE) |
|---|---|---|---|---|---|
| | Assumed risk with control | Assumed risk with peer-support | | | |
| **Retention in care** | | | | | |
| Retention in care (6 mo) | 64.6% | 67.1% | RR = 1.05 (0.92, 1.20) | 1916 (4 RCTs) | ⊕⊕⊕◯ MODERATE [1,2] |
| Retention in care (12 mo) | 74.8% | 80.0% | RR = 1.07 (1.02, 1.12) | 1556 (3 RCTs) | ⊕⊕⊕⊕ HIGH |
| **ART initiation** | | | | | |
| ART initiation (12 mo) | 24.0% | 18.8% | RR = 0.99 (0.74, 1.32) | 180 (2 RCTs) | ⊕⊕◯◯ LOW [1,3] |
| **ART adherence** | | | | | |
| ART adherence (3 mo) | 78.2% | 81.8% | RR = 1.06 (1.01,1.10) | 1282 (4 RCTs) | ⊕⊕⊕⊕ HIGH |
| ART adherence (6 mo) | 72.3% | 74.8% | RR = 1.03 (0.97, 1.08) | 1823 (6 RCTs) | ⊕⊕⊕◯ MODERATE [3,4] |
| ART adherence (12 mo) | 87.3% | 92.1% | RR = 1.05 (0.98, 1.12) | 1140 (4 RCTs) | ⊕⊕⊕◯ MODERATE [1] |
| Pill count adherence <95% (12 mo) | 2.4% | 1.4% | RR = 0.57 (0.23, 1.37) | 1336 (1 RCT) | ⊕⊕◯◯ LOW [5] |
| Pill count adherence <100% (12 mo) | 23.3% | 25.5% | RR = 1.09 (0.87, 1.37) | 1336 (1 RCT) | ⊕⊕⊕◯ MODERATE [3] |
| Any missed doses (12 mo) | 19.2% | 17.6% | RR = 0.91 (0.71, 1.19) | 1336 (1 RCT) | ⊕⊕⊕◯ MODERATE [3] |
| **CD4 cell count** | | | | | |
| CD4 cell count (6–24 mo) | Estimates shown in Table 2. No studies found a stat.sign. difference between the groups | | | 2733 (5 RCTs) | ⊕⊕◯◯ LOW [3,6] |
| **Viral load/suppression/failure** | | | | | |
| Viral suppression (3 mo) | RR = 0.98 (0.85, 1.14) to RR = 1.31 (1.03, 1.67). See Table 2. | | | 283 (2 RCTs) | ⊕⊕◯◯ LOW [1,3] |
| Viral suppression (6 mo) | RR = 1.02 (0.94, 1.11) to OR = 6.24 (1.28–30.5). See Fig 6 and Table 2. | | | 704 (7 RCTs) | ⊕⊕⊕⊕ HIGH |
| Viral suppression (12 mo) | RR = 0.80 (0.64, 0.99) to RR = 1.38 (1.03, 1.85). See Table 2. | | | 494 (2 RCTs) | ⊕⊕◯◯ LOW [1,3] |
| Virologic failure (6 mo) | 7.0% | 6.0% | RR = 0.93 (0.60, 1.45) | 1275 (2 RCTs) | ⊕⊕◯◯ LOW [3,6] |
| Virologic failure (12 mo) | 6.8% | 6.4% | RR = 0.79 (0.53, 1.19) | 1468 (3 RCTs) | ⊕⊕◯◯ LOW [3,6] |
| Virologic failure (18 mo) | 2.2% | 3.5% | RR = 1.23 (0.34, 4.45) | 1162 (2 RCTs) | ⊕◯◯◯ VERY LOW [5,6] |
| Virologic failure (24 mo) | 4.3% | 3.8% | RR = 0.54 (0.31, 0.94) | 1172 (2 RCTs) | ⊕⊕⊕◯ MODERATE [6] |
| **Quality of life** | | | | | |
| QoL–mental (12 mo) | Estimates shown in Fig 8. | | SMD = -0.06 (-0.27, 0.15) | 251 (2 RCTs) | ⊕⊕◯◯ LOW [1,3,4] |
| QoL–physical (12 mo) | Estimates shown in Fig 8. | | SMD = -0.04 (-0.35, 0.28) | 251 (2 RCTs) | ⊕⊕◯◯ LOW [1,3,4] |
| **Mental health** | | | | | |

(*Continued*)

**Table 3.** (Continued)

| | | | |
|---|---|---|---|
| Depressive symptoms (6–12 mo) | Estimates shown in Table 2. No studies found a stat.sign. difference between the groups | 338 (2 RCTs) | ⊕◯◯◯ VERY LOW [5,6] |

1. Downgraded by 1 level because of inconsistency.

2. 1 other RCT found a stat.sign. difference in favour of the intervention (see Table 2)

3. Downgraded by 1 level because of imprecision.

4. 1 other RCT found no stat.sign. difference between the groups (see Table 2)

5. Downgraded by 2 levels because of imprecision.

6. Downgraded by 1 level because of risk of bias.

Legend. CI: Confidence interval; RCT: Randomised controlled study; SD: Standard deviation.

*The risk in the intervention group (and its 95% confidence interval) is based on the assumed risk in the comparison group and the relative effect of the intervention (and its 95% CI).

The results of our systematic review are aligned with the WHO guidelines [2, 5], which state that peer-support can help people prepare and start therapy, and to previous reviews [4, 12–14, 18, 28], but they are novel in that they quantitatively demonstrate evidence of not only short-term but also important long-term effects, most notably retention in care. Long-term outcomes are critical with ART because it is a lifelong treatment. The evidence for peer-support of ART adherence is strongest in the short-term, but promising also up to one year follow-up, which has important implications because better ART adherence is related to disease suppression and reduced transmission risk. Importantly, peer-support offers some support for two of the Joint United Nations Programme on HIV and AIDS (UNAIDS) 90-90-90 targets [59], namely that diagnosed people are linked to care and are virally suppressed, which aligns with results from Dave and colleagues [28] for community-based initiatives.

Related, the observed variation in and tailoring to the community context of the peer-support intervention are important to note. Recall, we included only in-person, of at least one hour peer-support between individuals considered equal, consisting of assistance and encouragement in daily disease management, social and emotional support, and linkage to clinical care and/or community resources. Yet, there were considerable heterogeneity of intervention characteristics, delivery models, intensity, and theoretical foundations. A likely interpretation is that our results indicate that different variations of peer-support in different settings for different key populations appear effective. Presumably, this is because there is one or more common effective factors of the peer providing non-judgmental support, role-modeling, and personalized advice on daily activities and functioning. Our findings therefore support the Global Health Sector Strategy on HIV 2016–2020 [5], which emphasises the importance of HIV services adapted for different populations and locations.

The results in favour of peer-support for people living with HIV have relevance to policy makers, healthcare providers, and users. As noted, our results align with existing global strategies and guidelines [5, 59] on the use of peer-support for linkage to care and adherence, and thus indicates benefits of scale up of peer-support. This may be particularly important in low income countries where there are human resource- and financial shortages, and HIV services are scarce [60]. Peer-support can help shoulder existing services. Healthcare professionals should be made aware of this, so they can refer people living with HIV with adherence challenges. Improved medication adherence can potentially help millions of people to suppress the virus and increase their well-being. As indicated by also other reviews [4], Decroo and colleagues [61] asserted that expert patients were an untapped, but critical resource of ART provision in sub-Saharan Africa. In addition to being effective for linkage to care and adherence,

peer-support is also a type of care package that meets various people's needs and preferences in humane and holistic ways, it is aligned with patient preferences for HIV care [62], and is scalable and cost-effective [63]. Furthermore, studies on the lived experiences of peer-supporters showed that peers themselves appeared to be empowered by their function as experts in coping with a chronic disease. Supporting others like them seemed to have positive therapeutic outcomes, and allowed them to learn new skills, gain self-awareness, and become more visible in the community [64–66].

## Limitations and research implications

Our ability to conduct meta-analyses on the results from each trial was limited given inconsistent measurement and reporting. For most outcomes there were variations in metrics used. Even among the eight studies that measured viral suppression, results were reported as a mix of viral suppression/failure at various cut-points for detectability and/or changes in log viral load. As highlighted also by others [18, 28, 29], in order to allow for comparisons among studies, there is a need to standardize outcomes representing the stages along the HIV care continuum, harmonize operational cut-points for metrics used, and improve reporting of studies examining these outcomes. The heterogeneity of the studies, in particular outcomes and inconsistent measurements, limited our ability not only to pool study results, but also to conduct sensitivity analyses. As a result, it was neither possible to improve precision to any great extent, nor statistically assess potential differences across groups, such as key populations, geographical settings, or low vs high-income countries or peer-support with and without a theoretical framework. Careful inspection and consideration of the characteristics and results of studies, however, revealed no discernible patterns that would explain why some implementations of peer-support were more successful (e.g. [44, 53]) than others (e.g. [52]). Related, it is also important to note that a few of our meta-analyses had high statistical heterogeneity and these results must therefore be interpreted cautiously.

We note that less than half of the group differences were statistically significant and many were too small to be considered clinically significant on an individual level. Reasons for the null effects differed across the included trials, including inadequate power, and previous research has noted the challenges in testing HIV-related trials on biomedical outcomes due to methodological considerations [67]. In regard to key populations, all were represented in our included trials, except sex workers and their clients, and research on peer-support among this group is needed. Related, there was only one eligible trial with females only, and given that HIV prevalence rates are higher for women in low income countries [68], more research should be carried out on the effects of peer-support among women. There were only three studies that used peer-support to assess improvements in care among MSM, the number of transgender people was low and it would be beneficial with more studies focused on these most-at-risk populations, who are not only disproportionately affected by HIV, but have lower access to and coverage of HIV testing and ART [2, 4, 5]. While we recognize the difficulty in replicating peer-support studies in different settings due to the contextually specific nature of the intervention, it would be beneficial with further studies. Paradoxically, outcomes often highlighted as integral to peer-support–improving mental health, loneliness, stigma, social support [19, 20, 23, 25]–were rarely reported in our included studies, and more studies to address also these effects would give valuable information that could strengthen HIV policy and programs. It is possible such factors are important causes of ART non-adherence [16], and research into these links are warranted.

Lastly, for resource reasons, we only included studies published in English and Scandinavian languages, and it is possible, although we identified no such studies in our searches, that

eligible RCTs in other languages exist. It was beyond the scope of our review to assess cost-effectiveness, but a related scoping review identified only one cost analysis of peer-support for people living with HIV [21]. The analysis demonstrated that the yearly per patient costs of peer-support was about $8.74 and the cost to avert one virologic failure was $189. As the authors explained [63], these are reasonable costs in themselves, and, though more difficult to quantify, had task-shifting and costs of care for caregivers been considered, cost-effectiveness would be even greater. Further research examining costs and related value for organizations is needed.

Limitations notwithstanding, we believe that the greatest strength of our study is that it is the first systematic review with meta-analyses to examine the effects of in-person, peer-support for people living with HIV. Furthermore, we only included RCTs, of which most had low risk of bias. We conducted extensive searches and many included studies had null findings, suggesting also studies with unanticipated results were identified. Thus, although the limited number of studies with similar outcomes precluded an assessment of the potential risk of publication bias with funnel plot, it seems unlikely there is publication bias.

## Conclusions

This systematic review contributes to the growing body of literature on the effects of peer-support for people living with HIV by adding evidence relevant to key groups including healthcare users, providers, and policy makers of demonstrable impacts on retention in care, ART adherence, viral suppression, and virologic failure. The modest but important improvements in outcomes suggest that peer-support be considered as a treatment approach to support existing HIV care services for people living with HIV. Our findings support the need to standardize outcome measurements and reporting, test more interventions to address effects among the most at-risk populations, and evaluate outcomes like mental health, social support, and stigma in high disease burden, low-resource settings.

## Supporting information

**S1 Table. PRISMA 2009 checklist.** Description of reporting of the systematic review.
(DOCX)

**S2 Table. Risk of bias assessment.** Description of risk of bias assessment of each included study.
(DOCX)

**S1 File. Protocol for the systematic review.**
(DOCX)

**S2 File. Search strategy for medline.**
(DOCX)

## Acknowledgments

We are grateful to librarian Ellen Sejersted for her excellent support and recommendations during the search process and professor Mariann Fossum for her assistance in selecting studies.

## Author Contributions

**Conceptualization:** Rigmor C. Berg.

**Data curation:** Rigmor C. Berg, Anita Øgård-Repål.

**Formal analysis:** Rigmor C. Berg.

**Methodology:** Rigmor C. Berg, Samantha Page, Anita Øgård-Repål.

**Project administration:** Rigmor C. Berg.

**Supervision:** Rigmor C. Berg.

**Validation:** Rigmor C. Berg, Samantha Page, Anita Øgård-Repål.

**Writing – original draft:** Rigmor C. Berg.

**Writing – review & editing:** Rigmor C. Berg, Samantha Page, Anita Øgård-Repål.

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
