## [Decision Letter · Decision Letter 0]

23 Feb 2021

PONE-D-21-00126

Is peer-support for people living with HIV effective? A systematic review and meta-analysis

PLOS ONE

Dear Dr. Berg,

Thank you for submitting your manuscript to PLOS ONE. After careful consideration, we feel that it has merit but does not fully meet PLOS ONE’s publication criteria as it currently stands. Therefore, we invite you to submit a revised version of the manuscript that addresses the points raised during the review process.

We look forward to receiving your revised manuscript.

Kind regards,

Tim Mathes

Academic Editor

PLOS ONE

Journal Requirements:

2. Please attach a Supplemental file of the results of the individual components of the quality assessment, not just the overall score, for each study included. Please also explain the reasons, and number of studies excluded for each reason, in the flow diagram. Thank you.

3. Please provide any updates you might have since the original search was performed in May 2020, or please provide the rational for ending your search at that time.

5. Thank you for stating the following in the Financial Disclosure section:

We note that one or more of the authors are employed by a commercial company: PRAXIS, Kristiansand, Norway

Additional Editor Comments:

In case of high heterogeneity/statistical significant heterogeneity (e.g. adherence 6 and 12 months) no meta-analysis should be performed or heterogeneity should be explored (e.g. performing subgroup analyses) and plausibly explained why performing a meta-analysis is not misleading nevertheless.

Reviewers' comments:

Reviewer's Responses to Questions

**Comments to the Author**

1. Is the manuscript technically sound, and do the data support the conclusions?

Reviewer #1: Yes

Reviewer #2: Yes

2. Has the statistical analysis been performed appropriately and rigorously? 

Reviewer #1: Yes

Reviewer #2: Yes

3. Have the authors made all data underlying the findings in their manuscript fully available?

Reviewer #1: Yes

Reviewer #2: Yes

4. Is the manuscript presented in an intelligible fashion and written in standard English?

Reviewer #1: Yes

Reviewer #2: Yes

5. Review Comments to the Author

Reviewer #1: Overall:

-Very well-done review and meta-analysis with many important discussion points. I have minor suggestions overall and recommend publication.

Introduction:

- Only tiny changes:

1) add an ‘s’ to word ‘setting’ on line 68

2) ‘casted’ on line 117 should be ‘cast’

Methods:

- line 179 – notes that a list of excluded studies can be requested – while I don’t think providing the list is necessary, it would be helpful to summarize the main reasons for exclusion, and if possible include the numbers of excluded for each reason (this could be included in Figure 1 if easiest)

Results:

- I’m not seeing information in the text or Table 1 about which specific aspects of the included studies’ interventions included peer support. For example, what tasks did the peers typically take on as part of their roles? I think this will be of interest to readers and should be summarized somewhere, even briefly.

- I see Risk behaviors listed as an outcome 7 times in Table 1, but this is not included in your list of most commonly reported outcomes on line 262. Please adjust.

- line 366 - you mention the stigma outcome and reference [54], but I think you mean Wouters et al 2014 [55]? Also you say there was a number of ROBs for that reference, but I don’t see that as very evident in the ROB figure – there is more unclear rather than high risk necessarily (which is similar to some other studies, like Broadhead). It looks like only Fogarty had a number of high ROBs. Can you maybe explain the issues with that study a bit further, perhaps in the discussion? I’m wondering if they used a peculiar method of peer support to result in increased stigma.

- line 355 – mentions ‘adherence to care’ – was this a different outcome from retention in care?

Discussion:

-line 370 you note similar reviews as [4, 20, 26-28, 36, 56] but 36 does not appear to be a review

-I know of a few other reviews that look at peer support interventions for people living with HIV beyond the ones you cite (Boucher et al 2020, https://pubmed.ncbi.nlm.nih.gov/31598801/; Embuldeniya et al 2013, https://www.sciencedirect.com/science/article/pii/S0738399113000530?via%3Dihub), but of particular relevance may be Kanters et al 2016 which also includes a meta-analysis: https://onlinelibrary.wiley.com/doi/full/10.7448/IAS.19.1.21141 - Also it looks like some of your references to other reviews mention doing meta-analyses as well? Please ensure your text acknowledges how your meta-analysis relates to these others.

-I’m confused by this sentence on line 471: “The analysis demonstrated a yearly per patient costs of about $8.74 and $189 to avert one virologic failure.” – why are there 2 cost amounts but only 1 measure listed? Is it a range of costs for the measure? Please clarify.

-line 472 – ‘through’ should be ‘though’

-line 403 you say “our results indicate that different variations of peer support in different settings for different key populations appear effective. Presumably, this is because there is one or more common effective factors of the peer providing non-judgmental support, role-modeling, and personalized advice on daily activities and functioning.” – Following from this, I think it would be helpful to reiterate in the discussion the definition of peer support you used – e.g. minimum of 60 minutes face-to-face interaction etc. – because definitions vary widely. Also maybe some details from the studies if available (if not, it’s a common limitation). I think it is a strength and demonstrates the importance of personalized interaction (e.g. tailoring support to a person’s needs), which could also be useful to emphasize as recommendations for future peer support initiatives.

-conclusion doesn’t mention the improvement in viral suppression – would be good to add there too

Other:

- figures 7 and 9 – the “favours control” vs. “favours peer” x-axis labels order is flipped compared to all the other figures – should this be consistent with others?

- figure 8 – any reason for using green squares rather than blue?

Reviewer #2: The study by Berg et al., with the title " Is peer-support for people living with HIV effective? A systematic review and meta-analysis ", aims to investigate the effectiveness of peer-support intervention on HIV care continuum and the quality of life of PLWH. The study is sound and is suggested to be the first to investigate the effect of peer-support on PLWH. The main objective of the study is to synthesize evidence on the effectiveness of peer-support intervention in the HIV community. This study could be relevant to policies maker and help improve the management of healthcare and lives of people living with HIV. However, the study showed some limitations, partly due to the inconsistency of the RCTs used for meta-analysis. Please see my comments below on some of these weaknesses.

Comments:

• Please include the viral suppression failure in the abstract and discussion, as this was part of the meta-analysis.

Title:

• The author should consider revising the title to: " The effectiveness of peer-support for people living with HIV: A systematic review and meta-analysis".

• The research question should be included in the introduction with the objectives and outcomes.

Abstract:

• The abstract should follow the Preferred Reporting Items for Systematic Reviews and Meta-Analyses (PRISMA) and include the objectives, outcomes and inclusion/exclusion criteria of the study.

• Please include the viral suppression failure.

Introduction:

• Please include reference in the second sentence.

• Please change "Further" to "Furthermore", page 2, line 66.

• Please add change "setting" to "settings", page 2, line 68.

• Please add care to the following: "access to care", page 2, line 73.

• Please remove "e.g" and just go with the citation, page 3, line 84.

• Please correct to "Dennis et al.", page 3, line 98.

• Please cite the book, page 3, line 99.

• Please remove "e.g" and just go with the citation, page 4, line 113.

• The author should clearly highlight the difference between this study and previous studies ( Simoni and Genberg). The research question could be also stated here before the objectives. Clearly define the objectives, aims and anticipated outcomes of interest in the context of peer-support and this study.

Methods:

• Please change "Material and methods" to "Methods" or "Methods and Analyses"

• Please cite only, remove "as describe above", page 5, line 140.

• Please use a less stigmatizing expression such as "People living with HIV- PLWH" instead of "HIV positive person". Restructure the sentence to: "....to be given to PLWH by PLWH for a minimum....", page 5, line 141.

• Remove "(or viral suppression/failure)", page 5, line 146.

• Is adherence to ART a primary or secondary outcome?, page 5, line 147.

• Please remove "OVID" from all the databases, page 6, lines 157-160.

• Search strategy: Please provide an example of the key words or combination of words used for the search. Appendix 3 doesn't show a specific example of search strategy.

Study selection

• Please use Rayyan QCRI.

• Please add a section for Outcomes (primary and secondary outcomes).

Data analysis

• Please keep reference [35] and delete the rest of the sentence: "for more information.........org".

Results:

• Delete line 232, page 9.

• Delete line 269, page 14.

• Delete line 286- 292, page 14

• Please clarify this sentence: page 20, lines 304-305.

• RR=1.06, 95% CI (1.01,1.10) do not seem to be statistically significant. There seems to be no difference between both arms of the study, page 20, line 315. What was the P-value?

• Please define primary and secondary outcomes in the "methods" section.

• Please include the number of excluded studies in the PRISMA flow diagram (Figure 1).

• Although the RR and CI did not show strong evidence of the intervention effect, we seem to see a statistical significance with Retention in care (12 months), ART adherence (3 months) and viral suppression at 6 months (Figure 2, Enriquez and Graham, no meta- analysis) and virological failure (24 months), see Tables 2, 3, 5, and 7. Please discuss these findings more in the "discussion section". What does viral failure means in the context of peer-support?

Discussion

• Please include the CI and the study significance in the discussion, page 22, line 374-376. Also include the reference Table.

• Please discuss the viral failure as well.

• Please highlight the overall statistics of outcomes with significance.

• Please discuss the results based on the evidences. Most of these RCT didn't show difference in the outcome (no significance) after the trial, page 23, lines 382-385.

• Please include CI and P-value, page 23, line 396 to 398.

6. PLOS authors have the option to publish the peer review history of their article (what does this mean?). If published, this will include your full peer review and any attached files.

Reviewer #1: No

Reviewer #2: **Yes: **Pascal Djiadeu

---

## [Author Response · Author response to Decision Letter 0]

9 Apr 2021

See attached document 'response to reviewers'

---

## [Editor Report · Decision Letter 1]

16 Apr 2021

PONE-D-21-00126R1

The effectiveness of peer-support for people living with HIV: A systematic review and meta-analysis

PLOS ONE

Dear Dr. Berg,

Thank you for submitting your manuscript to PLOS ONE. After careful consideration, we feel that it has merit but does not fully meet PLOS ONE’s publication criteria as it currently stands. Therefore, we invite you to submit a revised version of the manuscript that addresses the points raised during the review process.

We look forward to receiving your revised manuscript.

Kind regards,

Tim Mathes

Academic Editor

PLOS ONE

Journal Requirements:

Additional Editor Comments (if provided):

Thank you for addressing heterogeneity in the meta-analyses. However, the explanations and rational for excluding studies from the analyses should be guided by clinical aspects rather than statistical reasons (e.g. "This was due to the large positive effect"), i.e. you should give possible clinical reasons why the effects of some studies differ. Excluding studies only because they cause statistical heterogeneity is not adequate.

The list of excluded studies should be provided as supplemental material. Alternatively, you might delete "and studies excluded after full-text consideration were listed.

---

## [Author Response · Author response to Decision Letter 1]

14 May 2021

Please see attached document 'response to reviewers'

---

## [Editor Report · Decision Letter 2]

19 May 2021

The effectiveness of peer-support for people living with HIV: A systematic review and meta-analysis

PONE-D-21-00126R2

Dear Dr. Berg,

We’re pleased to inform you that your manuscript has been judged scientifically suitable for publication and will be formally accepted for publication once it meets all outstanding technical requirements.

Kind regards,

Tim Mathes

Academic Editor

PLOS ONE
---

## [Editor Report · Acceptance letter]

24 May 2021

PONE-D-21-00126R2 

The effectiveness of peer-support for people living with HIV: A systematic review and meta-analysis 

Dear Dr. Berg:

I'm pleased to inform you that your manuscript has been deemed suitable for publication in PLOS ONE. Congratulations! Your manuscript is now with our production department. 

Kind regards, 

on behalf of

Dr. Tim Mathes 

Academic Editor

PLOS ONE